**Data Availability Statement:** All relevant data are within the manuscript and its Supporting Information files.

# Risk of chronic kidney disease in patients with heat injury: A nationwide longitudinal cohort study in Taiwan

**Min-Feng Tseng**[1,2,3], **Chu-Lin Chou**[3,4,5], **Chi-Hsiang Chung**[6,7], **Ying-Kai Chen**[2], **Wu-Chien Chien**[6☯], **Chia-Hsien Feng**[1,8☯], **Pauling Chu**[3,9]*

1 Ph.D. Program in Toxicology, College of Pharmacy, Kaohsiung Medical University, Kaohsiung, Taiwan, 2 Department of Internal Medicine, Zuoying Branch of Kaohsiung Armed Forces General Hospital, Kaohsiung, Taiwan, 3 Division of Nephrology, Department of Internal Medicine, Tri-Service General Hospital, National Defense Medical Center, Taipei, Taiwan, 4 Division of Nephrology, Department of Internal Medicine, Shuang Ho Hospital, Taipei Medical University, New Taipei City, Taiwan, 5 Division of Nephrology, Department of Internal Medicine, School of Medicine, College of Medicine, Taipei Medical University, Taipei, Taiwan, 6 School of Public Health, National Defense Medical Center, Taipei, Taiwan, 7 Taiwanese Injury Prevention and Safety Promotion Association, Taipei, Taiwan, 8 Department of Fragrance and Cosmetic Science, College of Pharmacy, Kaohsiung Medical University, Kaohsiung, Taiwan, 9 Center for the Prevention and Treatment of Heat Stroke, Tri-Service General Hospital, Taipei, Taiwan

☯ These authors contributed equally to this work.
* pauling.chu@gmail.com

## Abstract

Global climate change has led to a significant increase in temperature over the last century and has been associated with significant increases in the severity and frequency of heat injury (HI). The consequences of HI included dehydration and rhabdomyolysis, leading to acute kidney injury, which is now recognized as a clear risk factor for chronic kidney disease (CKD). We aimed to investigate the effects of HI on the risk of CKD. This nationwide longitudinal population-based retrospective cohort study utilized the Taiwan National Health Insurance Research Database (NHIRD) data. We enrolled patients with HI who were followed in NHIRD system between 2000 and 2013. We excluded patients diagnosed with CKD or genital-urinary system-related disease before the date of the new HI diagnosis. The control cohort consisted of individuals without HI history. The patients and control cohort were selected by 1:4 matching according to the following baseline variables: sex, age, index year, and comorbidities. The outcome measure was CKD diagnosis. In total, 815 patients diagnosed with HI were identified. During the 13 year observation period, we identified 72 CKD events (8.83%) in the heat stroke group and 143 (4.38%) CKD events in the control group. Patients with heat stroke had an increased risk of CKD than the control patients (adjusted HR = 4.346, P < 0.001) during the follow-up period. The risk of end-stage renal disease was also significantly increased in the heat stroke group than in the control group (adjusted hazards ratio: 9.078, p < 0.001). HI-related CKD may represent one of the first epidemics due to global warming. When compared to those without HI, patients with HI have an increased CKD risk.

**Funding:** The author(s) received no specific funding for this work.

**Competing interests:** The authors have declared that no competing interests exist.

# Introduction

Heat injury (HI) is the accumulation of heat resulting in the body's inability to tolerate it. Heat-related illness can range from mild conditions such as a vertigo, skin rash, or cramps to very serious conditions, such as heat syncope and heat exhaustion [1]. Heat stroke is the most severe heat-related illness and the characteristic of heatstroke is body temperature >40˚C combined with neurologic dysfunction. Heat stroke is a type of severe heat illness with life-threatening injury requiring emergent and intensive care, and it accounts for 600 deaths a year in the United States [2]. In addition, the 28-day mortality rate of heat stroke has been reported to reach up to 58% [3]. Global climate change has led to a significant increase in temperature over the last century and has been associated with significant increases in the severity and frequency of HI.

Acute kidney injury (AKI) caused by HI is often combined with rhabdomyolysis. Rhabdomyolysis is a clinical and biochemical syndrome that occurs when the skeletal muscle cells disrupt and release creatine phosphokinase, lactate dehydrogenase, and myoglobin into the interstitial space and plasma. AKI occurs in 33%-50% of patients with rhabdomyolysis. Its etiology is multifactorial, which includes renal intraluminal cast formation, vasoconstriction, and direct myoglobin toxicity [4]. However, sustained AKI can lead to renal interstitial fibrosis, reductions in nephron number, insufficient blood supply, cell cycle disruption, and disrupted repair mechanisms, which eventually cause chronic kidney disease (CKD). In turn, CKD is also a risk factor of AKI development. Both diseases have a close relationship, are associated with an increased risk of nephron death, and can cause serious sequelae such as end-stage renal disease (ESRD) [5] However, it is unclear whether heat injury causes long-term renal damage.

Herein, we aimed to evaluate the relationship between HI and CKD using the National Health Insurance Research Database (NHIRD) of Taiwan. In addition to HI, other systemic co-morbidities were also examined in the multivariate analysis model to investigate whether HI is an independent risk factor for CKD.

# Materials and methods

## Data source

This retrospective population-based cohort study was approved by the Institutional Review Board (IRB) of Tri-Service General Hospital (IRB Registration Number: 2-105-05-082). All data are fully anonymized before we collected them and the IRB had waived the requirement for informed consent.

This study confirmed that all experiments were performed by relevant guidelines and regulations. The study enrolled all patients diagnosed with HI (ICD-9-CM 992.X) in Taiwan. Data from the NHIRD in Taiwan were used in this study [6]. The association between heat stroke and CKD events was investigated between 2000 to 2013 period. More than 99% of the population in Taiwan was covered by the National Health Insurance Program. The international Classification of Diseases, Ninth Revision (ICD-9) code was used for diagnosis [7]. The database in NHIRD also includes medications and patients' demographics (such as socioeconomic status and residential area).

## Patient selection

Patients diagnosed with HI from 2000 to 2013 were enrolled in the study group. To further enhance the diagnostic accuracy in the study group, only patients with the abovementioned

ICD-9 code obtained from the emergency room (department code: 02) were included to ensure that only the patients with HI diagnosis were included.

In addition, patients were excluded if any of the criteria are present to remove any confounding factors: (1) received any renal surgery in the study period, (2) diagnose with CKD (ICD-9 code: 585) before the index date, and (3) presence of acute renal failure (ICD-9 code: 584.5), any genitourinary tract functional and infectious disease (ICD-9 codes: 593.81, 594.1, 596.0, 596.4, 596.5, 344.61, 596.5, 596.8, 597.80, 598.9, and 599.0), anomalies of the genital organ (ICD-9 codes: 752.9, 753.10,753.12), obstructive nephropathy (ICD-9 codes: 599.6, 591, 592.0, 592.1, 592.9, 593.3, 593.7), glomerulonephritis (ICD-9 codes: 580.9, 581.0, 582.1, 582.2, 582.4, 582.9, 583.4 583.4, 583.89, 583.9), and nephrotic syndrome (ICD-9 code: 581.X); and (5) age <18 years or >100 years.

The individuals in the study group were age-, sex-, and comorbidity-matched to a non-HI individual at a ratio of 1:4, who served as the control group. HI patients who could not be matched to non-HI individuals were excluded.

## Main outcome measurement

The primary outcome in the current study was the development of CKD, which was represented with the ICD-9 code of 585 after the index date (Jan 1, 2000).

Individuals were identified as having CKD if they had a diagnosis should be confirmed for at least 3 consecutive times at intervals of at least 3 months according to ICD-9-CM code 585 (CKD); thus, the possibility to misdiagnose CKD is minimal. These CKD patients neither had other kidney-related conditions nor had received renal dialysis or a transplant before the cohort entry date. Thus, all patients had a primary diagnosis of CKD.

Patients with CKD were divided into subgroups according to their eGFR, which was calculated according to MDRD (modification of diet in renal disease study) equation [8]. Moreover, we also analyzed the CKD subgroup requiring renal replacement therapy. The renal replacement therapy subgroup enrolled CKD patients requiring hemodialysis or peritoneal dialysis for >3 months.

To increase the accuracy, we considered the effect of demographic conditions (i.e., age, sex, urbanization, and level of care) and the following comorbidity to standardize the baseline status in the study population: hypertension (ICD-9 codes: 401–405), hyperlipidemia (ICD-9 codes: 272.0–272.9), diabetes mellitus (DM) (ICD-9 codes: 250.x), cerebrovascular disease (ICD-9 codes: 362.34, 430.x–438.x), and congestive heart failure (ICD-9 codes: 398.91, 402.01, 402.11, 402.91, 404.01, 404.03, 404.11, 404.13, 404.91, 404.93, 425.4–425.9, 428.x).

We longitudinally traced the data from the index date until the date of CKD diagnosis between January 1, 2000 and December 31, 2013, withdrawal from the national health insurance program, death, or on 31 December 2013.

## Statistical analysis

The baseline demographic condition and comorbidities were compared between the study and control groups using descriptive statistics. Chi-square test were used for categorical variables and *t*-test for continuous variables [9].

The two groups were adjusted carefully with respect to known confounders to ensure comparability during analyses. Multivariate Cox models were used simultaneously for age, sex, hypertension, hyperlipidemia, diabetes mellitus, heat failure, season, location, urbanization level, and level of care.

The Kaplan-Meier analysis was applied for the cumulative incidence curves of CKD for the two cohorts, and differences between cohorts were evaluated using the log-rank test. The

incidence of CKD was estimated based on the Poisson distribution, with 95% confidence intervals (CIs). The odds ratios of CKD incidence rate after HI in the univariate analysis was used for the conditional logistic regression analyses.

In the outcomes analysis, $P < 0.05$ were interpreted as statistically significant. The type I error due to multiple testing was corrected by the Bonferroni method. The P value $< 0.001$ was considered to be significant for multiple comparisons. All data analyses were conducted using SPSS software version 22 (SPSS Inc., Chicago, IL, USA).

## Results

In this study, a total of 1146 individuals diagnosed with heat stroke for the first time were enrolled. Of these, 331 patients with heat stroke were excluded (including heat stroke before index date, CKD before tracking, genitourinary system-related disease, age <18 years, unknown sex). After applying the exclusion criteria and four propensity score matching by sex, age, index date, and comorbidities. a total of 815 patients and 3260 matched controls were enrolled, as shown in Fig 1. During the 13-year follow-up, we identified 72 CKD events (8.83%) in the heat stroke group and 143 (4.38%) CKD events in the control group.

In Table 1, the baseline characteristics of the patients and controls were comparable in terms of sex, age, and comorbidity (hypertension, hyperlipidemia, diabetes mellitus, and stroke). Compared to patients without heat stroke, patients with heat stroke were more likely

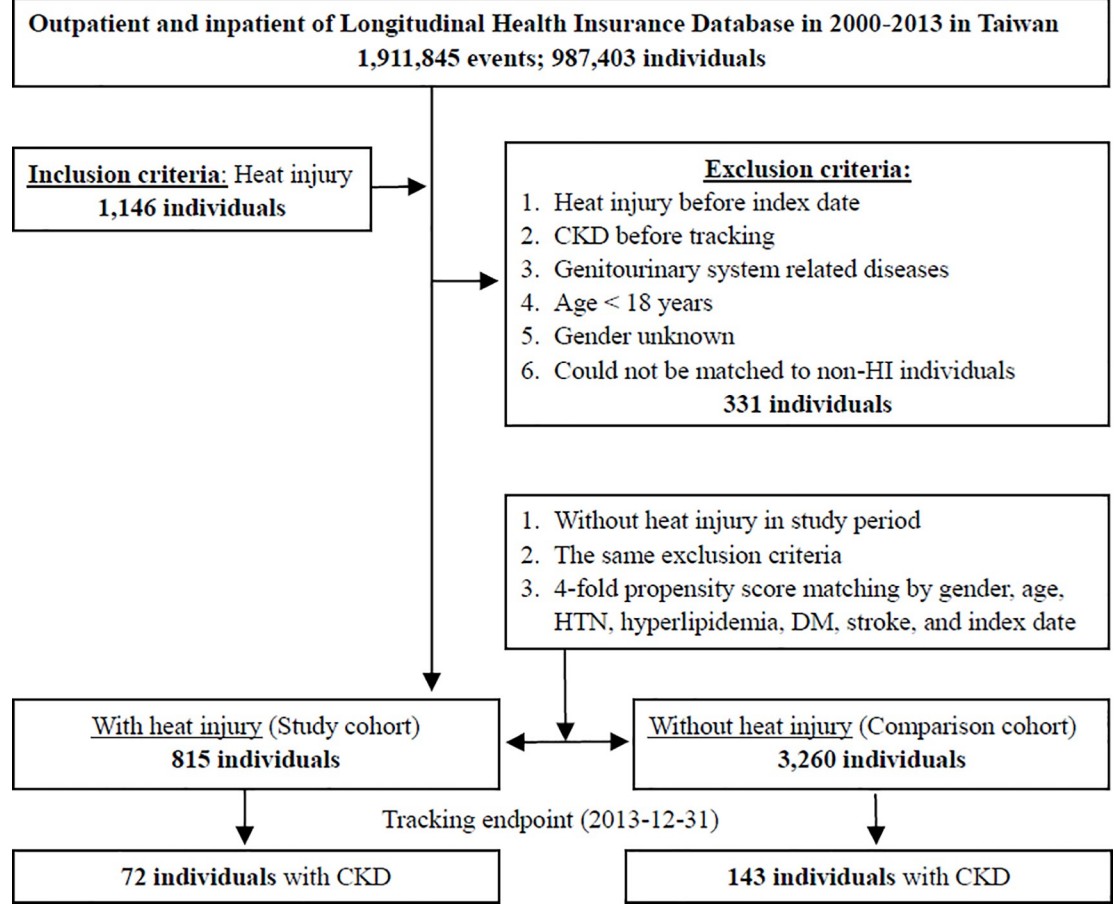

**Fig 1. Patient selection flow chart.** CKD: Chronic kidney disease.

**Table 1. Characteristics of study in the baseline.**

| Heat injury | Total | | With | | Without | | P |
|---|---|---|---|---|---|---|---|
| Variables | n | % | n | % | n | % | |
| Total | 4,075 | 100 | 815 | 20 | 3,260 | 80 | |
| Gender | | | | | | | 0.999 |
| Male | 3,345 | 82.09 | 669 | 82.09 | 2,676 | 82.09 | |
| Female | 730 | 17.91 | 146 | 17.91 | 584 | 17.91 | |
| Age (years) | 43.12 ± 18.40 | | 42.74 ± 21.16 | | 43.21 ± 17.64 | | 0.514 |
| HTN | | | | | | | 0.159 |
| Without | 3,683 | 72.56 | 726 | 89.08 | 2,957 | 90.71 | |
| With | 392 | 27.44 | 89 | 10.92 | 303 | 9.29 | |
| Hyperlipidemia | | | | | | | 0.517 |
| Without | 3,980 | 78.06 | 799 | 98.04 | 3,181 | 97.58 | |
| With | 95 | 21.94 | 16 | 1.96 | 79 | 2.42 | |
| DM | | | | | | | 0.274 |
| Without | 3,531 | 69.08 | 716 | 87.85 | 2,815 | 86.35 | |
| With | 544 | 30.92 | 99 | 12.15 | 445 | 13.65 | |
| Stroke | | | | | | | 0.762 |
| Without | 3,992 | 76.37 | 780 | 95.71 | 3,112 | 95.46 | |
| With | 183 | 23.63 | 35 | 4.29 | 148 | 4.54 | |
| HF | | | | | | | 0.999 |
| Without | 4.225 | 79.02 | 805 | 98.77 | 3,220 | 98.77 | |
| With | 50 | 20.98 | 10 | 1.23 | 40 | 1.23 | |
| Season | | | | | | | 0.714 |
| Spring | 531 | 13.03 | 106 | 13.01 | 425 | 13.04 | |
| Summer | 2,876 | 70.58 | 580 | 71.17 | 2,296 | 70.43 | |
| Autumn | 491 | 12.05 | 90 | 11.04 | 401 | 12.30 | |
| Winter | 177 | 4.34 | 39 | 4.78 | 138 | 4.23 | |
| Location in Taiwan | | | | | | | <0.001* |
| Northern | 1,292 | 31.71 | 103 | 12.64 | 1,189 | 36.47 | |
| Middle | 1,216 | 29.84 | 318 | 39.02 | 898 | 27.55 | |
| Southern | 1,073 | 26.33 | 156 | 19.14 | 917 | 28.13 | |
| Eastern | 432 | 10.60 | 207 | 25.40 | 225 | 6.90 | |
| Outlets islands | 62 | 1.52 | 31 | 3.80 | 31 | 0.95 | |
| Urbanization level | | | | | | | <0.001* |
| 1 | 618 | 15.17 | 39 | 4.79 | 579 | 17.76 | |
| 2 | 1,660 | 40.74 | 361 | 44.29 | 1,299 | 39.85 | |
| 3 | 584 | 14.33 | 98 | 12.02 | 486 | 14.91 | |
| 4 | 1,213 | 29.77 | 317 | 38.90 | 896 | 27.48 | |
| Level of care | | | | | | | 0.014 |
| Medical center | 1,085 | 26.63 | 246 | 30.18 | 839 | 25.74 | |
| Regional hospital | 1,519 | 37.28 | 304 | 37.30 | 1,215 | 37.27 | |
| Local hospital | 1,471 | 36.10 | 265 | 32.52 | 1,206 | 36.99 | |

P-value (category variable: Chi-square/Fisher exact test; continuous variable: t-test)

HT = Hypertension; DM = Diabetes mellitus; HF = Heart failure.

*: The p value was obtained after Bonferroni correction.

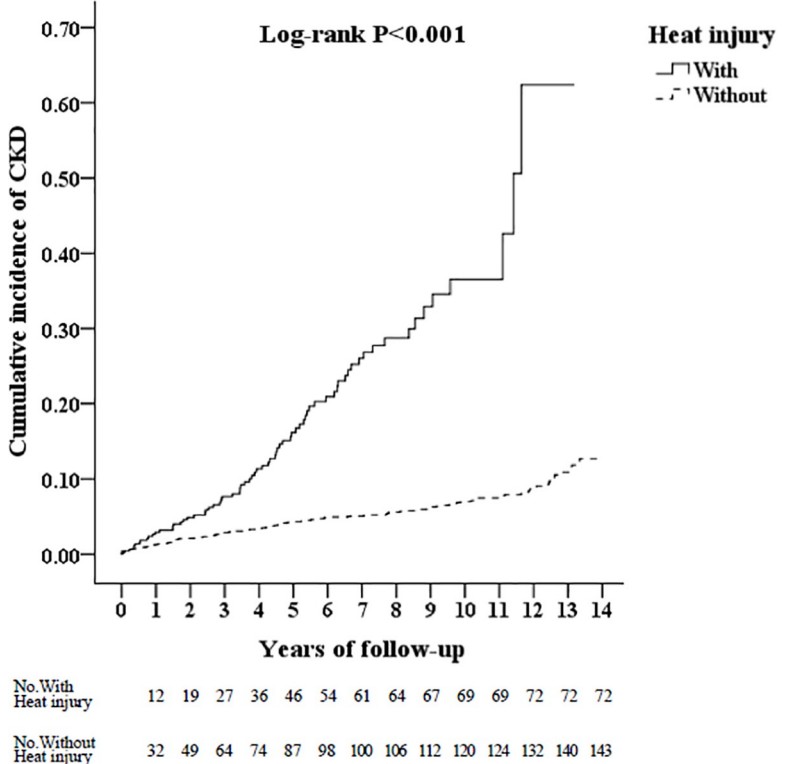

**Fig 2. Kaplan-Meier curve for cumulative risk of chronic kidney disease in patients with and without heat injury.**
CKD: Chronic kidney disease.

to live in middle or eastern Taiwan and outlet islands and in an area with lower urbanization level.

In Fig 2, a Kaplan–Meier curve for cumulative CKD risk stratified by heat stroke with a log-rank test was shown. Patients with heat stroke were associated with a significantly increased risk of CKD events (log-rank P < 0.001). The incidence of CKD events was higher in the heat stroke group than in the control group from the first year of follow-up to the 13th year.

The adjusted hazards ratio (HR) of CKD events in the subgroup of cases with heat stroke and the matched controls are depicted in Table 2. Patients with heat stroke had higher CKD events than the controls (adjusted HR = 4.346, $P < 0.001$) during the follow-up period after adjusting for age, sex, insurance premium, comorbidities, urbanization level, patient care quality, and residential area in Taiwan. The risks of CKD events were associated with comorbidities (hypertension, hyperlipidemia, diabetes, and heart failure). There was a close relationship found between heat stroke episodes and CKD events.

We also investigated the risk of ESRD in the heat stroke and matched groups. We found that the proportion of patients with ESRD was significantly increased in the heat stroke group than in the matched group (Table 3) (adjusted HR: 9.078, p <0.001).

Furthermore, the effect of HI on the CKD subgroups were examined according to ICD 9 codes in Table 4. The effect of HI was statistically significant in those CKD stage 2–5 and CKD of unknown stage group. HI patients compared with control group showed increased CKD stage 2 risk (95% CI = 1.038–2.014, adjusted HR: 1.432, P = 0.029), CKD stage 3 risk (95% CI = 4.232–8.496, adjusted HR: 5.265, P < 0.001), CKD stage 4 risk (95% CI = 5.998–11.225, adjusted HR: 7.984, P < 0.001), and CKD stage 5 risk (95% CI = 9.137–20.986, adjusted HR: 11.106, P < 0.001), respectively.

**Table 2. Hazard ratio of chronic kidney disease in association with baseline characteristics among heat injury patients in Cox model with competing risks.**

| Variables | Adjusted HR | 95% CI | 95% CI | P |
|---|---|---|---|---|
| **Heat injury** | | | | |
| **Without** | Reference | | | |
| **With** | 4.346 | 3.206 | 5.892 | <0.001* |
| **Gender** | | | | |
| **Male** | 1.187 | 0.864 | 1.630 | 0.290 |
| **Female** | Reference | | | |
| **Age (years)** | 1.024 | 1.015 | 1.033 | <0.001* |
| **HTN** | | | | |
| Without | Reference | | | |
| With | 1.615 | 1.421 | 1.897 | 0.012* |
| **Hyperlipidemia** | | | | |
| Without | Reference | | | |
| With | 1.181 | 1.044 | 1.736 | 0.017* |
| **DM** | | | | |
| Without | Reference | | | |
| With | 2.335 | 1.762 | 3.095 | <0.001* |
| **Stroke** | | | | |
| Without | Reference | | | |
| With | 1.043 | 0.620 | 1.754 | 0.875 |
| **HF** | | | | |
| Without | Reference | | | |
| With | 2.859 | 1.905 | 4.289 | <0.001* |
| **Season** | | | | |
| Spring | Reference | | | |
| Summer | 0.936 | 0.627 | 1.397 | 0.745 |
| Autumn | 0.922 | 0.632 | 1.346 | 0.675 |
| Winter | 1.109 | 0.754 | 1.629 | 0.599 |
| **Location** | | | | |
| Northern Taiwan | Reference | | | |
| Middle Taiwan | 1.031 | 0.720 | 1.478 | 0.866 |
| Southern Taiwan | 1.173 | 0.825 | 1.667 | 0.376 |
| Eastern Taiwan | 1.776 | 1.165 | 2.707 | 0.008 |
| Outlets islands | 1.697 | 0.415 | 6.939 | 0.462 |
| **Urbanization level** | | | | |
| 1 (The highest) | 1.131 | 0.693 | 1.365 | 0.465 |
| 2 | 1.094 | 0.506 | 1.427 | 0.974 |
| 3 | 1.058 | 0.487 | 1.513 | 0.597 |
| 4 (The lowest) | Reference | | | |
| **Level of care** | | | | |
| Medical center | 1.065 | 0.675 | 1.684 | 0.787 |
| Regional hospital | 1.008 | 0.724 | 1.388 | 0.992 |
| Local hospital | Reference | | | |

HR = hazard ratio, CI = confidence interval, Adjusted HR: Adjusted variables listed in the table

HT = Hypertension; DM = Diabetes mellitus; HF = Heart failure

*denotes P < .05 and was considered statistically significant.

**Table 3. Risk of ESRD receiving HD in patients with CKD by using Cox regression.**

| Heat injury | With HI vs. without HI(Reference) | | | | |
|---|---|---|---|---|---|
| | Events | Ratio | Adjusted HR | 95%CI | P |
| Without HD | 11 | 0.551 | 1.111 | 0.820–1.518 | 0.272 |
| With HD | 61 | 4.477 | 9.078 | 6.684–12.295 | <0.001* |

PYs = Person-years; Adjusted HR = Adjusted Hazard ratio: Adjusted for the variables listed in Table 2.; CI = confidence interval; ESRD = End stage renal disease; CKD = chronic kidney disease; HD = Hemodialysis; with HD represents heat injury patients with CKD and HD; without HD represents heat injury patients with CKD but without HD; ratio represents the risk of ESRD in HI vs non-HI.

In S1 and S2 Tables, the mean follow-up time of the heat stroke and control groups were 10.40 ± 13.70 and 10.97 ± 9.84 years, respectively. The average time between HI and onset of CKD was 4.23 ± 2.87 years. In contrast, the average time in the matched group was 4.82 ± 4.12 years.

## Discussion

In our retrospective population-based study of Taiwanese adults, patients with HI had a four-fold increased risk of CKD. Comorbidities including hypertension, hyperlipidemia, diabetes, and heart failure were also associated with higher incidence of CKD. Heat exhaustion is the most common HI during our 13-year follow-up period. HI was also associated with shorter progression duration to CKD than non-HI.

As the body temperatures increases, the occurrence of HI-associated complications, including central nervous system dysfunction and additional organ damage including AKI, liver injury, and rhabdomyolysis, is gradually increasing [10]. Recently, a population-based study enrolling 628 patients in Taiwan suggested that ischemic heart disease was independently associated with heat stroke in a Cox multivariate regression analysis [11]. Another retrospective study enrolled patients aged >65 years from 114 cities in USA and revealed that the impact of high temperature and heat waves increased the hospitalizations for renal disease (including acute/chronic glomerulonephritis, nephrotic syndrome, acute/chronic renal failure) and respiratory diseases (including bronchiectasis and chronic airway obstruction) [12]. A case-crossover study enrolling 19.17 million patients from New York, USA also demonstrated that high temperatures increased hospitalizations for acute renal failure, urinary tract infections, renal calculi, lower urinary calculi, and other lower urinary tract disorders [13]. Another time-series analysis concluded that high temperatures were associated with increases in morbidity and the relative risks of total emergency room visits and non-external hospitalizations [14].

**Table 4. Risk of different stages of CKD by using Cox regression.**

| Heat injury | With HI vs without HI (reference) | | | | |
|---|---|---|---|---|---|
| CKD stage | Events | Ratio | Adjusted HR | 95%CI | P |
| I | 3 | 0.387 | 0.787 | 0.511–1.098 | 0.485 |
| II | 7 | 0.710 | 1.432 | 1.038–2.014 | 0.029 |
| III | 22 | 3.021 | 5.265 | 4.232–8.496 | <0.001* |
| IV | 19 | 4.044 | 7.984 | 5.998–11.225 | <0.001* |
| V | 18 | 6.386 | 11.106 | 9.137–20.986 | <0.001* |
| Unknown | 3 | 2.554 | 5.134 | 3.726–7.133 | <0.001* |

Abbreviations: Adjusted HR = Adjusted Hazard ratio: Adjusted for the variables listed in Table 2.; CI: Confidence interval; CKD: Chronic kidney disease; HD: Hemodialysis

Although the exact mechanism of how heat injury can cause CKD is unknown, there are several possibilities. The ability of the thermoregulation relies on the cutaneous vasodilatation and sweat gland excretion to cool the surface of the skin, thereby reducing body temperature. Moreover, heat-induced peripheral vasodilation and dehydration may involve decreased intestinal and solid organ blood supply, which leads to ischemia. Several studies have focused on exercise and showed that heat stresses can damage the gut structure and change its permeability. A malfunctioned intestinal tight junction barrier allows increased permeation of bacteria and endotoxins into the blood circulation [15]. Additionally, decreased blood flow through the renal artery can cause permanent damage to the kidney tissues and can increase the risk of acute or chronic renal failure.

A recent study has shown that heat stress induced the increase in plasma lipopolysaccharide concentration, anti-inflammatory cytokines (IL-10 and IL-1ra) and inflammatory responsive cytokines (IL-6) [16]. The heat stress response to endotoxemia and systemic inflammation is similar to sepsis, which causes profound alteration of the macro- and microcirculation of the kidney and maldistribution of blood supply to the organs. These dramatic changes cause a significant decrease in renal functional capillary circulation and induce renal ischemia [17]. A review article identified that patients with AKI had higher risks of developing CKD, ESRD, and mortality than those without AKI. AKI was an independent risk factor for CKD and ESRD [18].

In this study, we also found that HI increased the risk of CKD, especially in individuals with comorbidities (hypertension, hyperlipidemia, diabetes, and heart failure) and those living in the eastern area of Taiwan. A cross-sectional survey including 23,869 participants shows that diabetes, hypertension, and dyslipidemia are risk factors of CKD [19].

Furthermore, compared to healthy people, individuals with comorbidities (hypertension, hyperlipidemia, diabetes, and heart failure) who experience heat stress generally have lower cardiac output and cutaneous blood flow [20]; thus their ability to supply renal blood perfusion and radiate heat from the skin are low. As a result, the core body temperature increases easily, which may lead to prolonged renal ischemia and increased risk of CKD and ESRD. This result is consistent with the result we found that HI is strongly associated with the increased severity of CKD.

The results of our study should be interpreted in light of its limitations and strengths. To overcome the confounding bias, we utilized a propensity score from the baseline population to match the diversity in characteristics between heat stroke and control groups. Residual confounding bias such as lifestyle factors, including body mass index, smoking, alcohol drinking, and medication compliance was poorly measured in the NHIRD database. We adjusted these factors by including related diseases such as hypertension, hyperlipidemia, diabetes mellitus, and stroke. Another limitation of our study was that we defined heat stroke using the ICD-9-CM codes. We cannot evaluate the cooling process, presence of air conditioning, and the dynamic change in the core body temperature during heat stroke. In the future, we might perform a longitudinal study to evaluate our study outcomes. The strengths of our retrospective study included the national database derived from one million sampled cases, the use of propensity score matching between the cases and controls, and application of case-controlled matched index date to ensure comparability during analyses and minimize the confounders' bias.

## Conclusions

We found that heat stroke was associated with an approximately four-fold increase in risk of CKD and nine-fold increase in the risk of ESRD requiring long-term renal replacement

therapy. Our results also demonstrated that the time interval to CKD progression decreases in heat stroke patients. Clinicians should continue to be alert for the appearance of CKD in high-risk patients with heat stroke.

## Supporting information

**S1 Table. Years of follow-up.**
(DOCX)

**S2 Table. Years to CKD.**
(DOCX)

## Acknowledgments

This study is supported by tri-service general hospital research grants TSGH-B-109010.

Role of the Funder/Sponsor: The funding institutions had no role in the design and conceptualization of the study; data collection /management/ analysis and interpretation; review or approval of the manuscript.

## Author Contributions

**Conceptualization:** Min-Feng Tseng, Wu-Chien Chien.

**Data curation:** Chi-Hsiang Chung, Wu-Chien Chien.

**Formal analysis:** Min-Feng Tseng, Chi-Hsiang Chung.

**Investigation:** Min-Feng Tseng, Chi-Hsiang Chung, Wu-Chien Chien.

**Methodology:** Min-Feng Tseng, Chi-Hsiang Chung, Wu-Chien Chien.

**Project administration:** Min-Feng Tseng, Chu-Lin Chou.

**Resources:** Min-Feng Tseng, Chu-Lin Chou, Chi-Hsiang Chung.

**Software:** Chu-Lin Chou, Ying-Kai Chen, Wu-Chien Chien.

**Supervision:** Ying-Kai Chen, Chia-Hsien Feng.

**Validation:** Min-Feng Tseng, Chia-Hsien Feng.

**Visualization:** Min-Feng Tseng, Chia-Hsien Feng.

**Writing – original draft:** Min-Feng Tseng, Chia-Hsien Feng, Pauling Chu.

**Writing – review & editing:** Min-Feng Tseng, Chia-Hsien Feng, Pauling Chu.

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
