## [Decision Letter · Decision Letter 0]

6 Jan 2020

PONE-D-19-33292

Risk of chronic kidney disease in patients with heat injury: A nationwide longitudinal cohort study in Taiwan

PLOS ONE

Dear Dr Chu,

Thank you for submitting your manuscript to PLOS ONE. After careful consideration, we feel that it has merit but does not fully meet PLOS ONE’s publication criteria as it currently stands. Therefore, we invite you to submit a revised version of the manuscript that addresses the points raised during the review process.

The paper from Tseng et al entitled "Risk of chronic kidney disease in patients with heat injury: A nationwide longitudinal cohort study in Taiwan" is an interesting paper on a  nationwide longitudinal cohort study in Taiwan. The cohort was  followed up during 13 years. The authors showed that patients with heat stroke had twice higher CKD events than in the control group and that the risk of end-stage renal disease was also significantly increased in the heat stroke group.

Both reviewers asked for Major revision.

Mainly, the authors did not define the different CKD stages of the patients (from 1 to 5 in the Patient selection part in Material and method. The stages should be defined with an explanation of the method used to classify patients. We think this can improve the results of the study.

The authors should also follow recommendations of Reviewer 2 for statistical analysis.

Some minor modifications should also be made in order to answer reviewers.

We would appreciate receiving your revised manuscript by 9th april 2020. To enhance the reproducibility of your results, we recommend that if applicable you deposit your laboratory protocols in protocols.io, where a protocol can be assigned its own identifier (DOI) such that it can be cited independently in the future. For instructions see: http://journals.plos.org/plosone/s/submission-guidelines#loc-laboratory-protocols

We look forward to receiving your revised manuscript.

Kind regards,

Valérie Metzinger-Le Meuth, PhD

Academic Editor

PLOS ONE

Additional Editor Comments:

The paper from Tseng et al entitled "Risk of chronic kidney disease in patients with heat injury: A nationwide longitudinal cohort study in Taiwan" is an interesting paper on a nationwide longitudinal cohort study in Taiwan. The cohort was followed up during 13 years. The authors showed that patients with heat stroke had twice higher CKD events than in the control group and that the risk of end-stage renal disease was also significantly increased in the heat stroke group.

Both reviewers asked for Major revision.

Mainly, the authors did not define the different CKD stages of the patients (from 1 to 5 in the Patient selection part in Material and method. The stages should be defined with an explanation of the method used to classify patients. We think this can improve the results of the study.

The authors should also follow recommendations of Reviewer 2 for statistical analysis.

Some minor modifications should also be made in order to answer reviewers.

2. In the ethics statement in the manuscript and in the online submission form, please provide additional information about the patient records/samples used in your retrospective study.

Specifically, please ensure that you have discussed whether all data/samples were fully anonymized before you accessed them and/or whether the IRB or ethics committee waived the requirement for informed consent.

If patients provided informed written consent to have data/samples from their medical records used in research, please include this information.

3. We noticed you have some minor occurrence(s) of overlapping text with the following previous publication(s), which needs to be addressed:

https://doi.org/10.1016/j.ejim.2018.09.019

https://doi.org/10.3390/ijerph16162865

https://doi.org/10.1016/j.radonc.2017.04.025

http://www.sjkdt.org/text.asp?2008/19/5/721/42439

In your revision ensure you cite all your sources (including your own works), and quote or rephrase any duplicated text outside the Methods section. Further consideration is dependent on these concerns being addressed.

Reviewers' comments:

Reviewer's Responses to Questions

**Comments to the Author**

1. Is the manuscript technically sound, and do the data support the conclusions?

Reviewer #1: Yes

Reviewer #2: Yes

2. Has the statistical analysis been performed appropriately and rigorously? 

Reviewer #1: Yes

Reviewer #2: Yes

3. Have the authors made all data underlying the findings in their manuscript fully available?

Reviewer #1: Yes

Reviewer #2: Yes

4. Is the manuscript presented in an intelligible fashion and written in standard English?

Reviewer #1: Yes

Reviewer #2: Yes

5. Review Comments to the Author

Reviewer #1: Tseng et al submit an original research paper entitled "Risk of chronic kidney disease in patients with heat injury: A nationwide longitudinal cohort study in Taiwan" In this work, they study the association between exposure to heat injury and onset of chronic kidney disease. Their longitudinal population-based retrospective cohort study used the Taiwan National Health Research Database data.

Authors selected adult patients diagnosed with HI that were followed up between 2000 and 2013. The outcome measure was CKD diagnosis. In total, 815 patients were diagnosed with HI, of which 72 were diagnosed with CKD. In total, that was twice more CKD diagnosis (in percentage) in the heat stroke group compared to the control group. The risk of end-stage

renal disease was also significantly increased in the heat stroke group.

The study is interesting and confirms in the Taiwanese population what was proposed by other studies, namely that

HI-related CKD may represent one of the first epidemics due to global warming.

The main drawback of the paper is that CKD diagnosis is very poorly defined; more informations have to be given.

Was CKD diagnosed according to eGFR? If yes, which equation was used to calculate it (CKD-EPI or other)?

The paper would gain a lot of relevance if the associations were to be performed again according to the various CKD stages, (1-5) or at least by subdividing patients in two groups , one for CKD stage1 to 3a, the other from CKD stage 3b to 5.

In figure 2, there seems to be an acceleration of CKD cases in the last three years in the HI group, can you please comment (heatwave in Taiwan or other explanation?)

Minor

Abstract

please correct "newly diagnosed with HI and were followed"

Introduction

Heat is repeated three times in "Heat injury (HI) is the accumulation of heat resulting in the body's inability to tolerate heat."

rephrase

reductions in nephron number, insufficient blood supply, cell cycle disruption, and disrupted repair mechanisms, which eventually CAUSE chronic kidney disease (CKD).

Rephrase sentence "In addition to HI, other systemic co-morbidities were also examined in the multivariate analysis model to investigate whether HI is an independent risk factor for CKD, which develops in the majority of the population."

Separate "Materials and methodsData source"

Results

separate "andcomorbidity ,"

Several word corrections still appear in the text, for example "The risk of CKD is was" and "in the heat stroke and matched groups," please correct

Please correct "Patients with heat stoke had higher CKD events"

Discussion rephrase ( HAs and repetition if increase) "A recent study HAS shown that heat stress induced the increase in plasma

lipopolysaccharide concentration and significant increases"

"which are in concordance with the findings in previous studies". Please cite them here

Reviewer #2: Kindly put in the LINE numbers so it facilitates identification of the lines that the reviewer needs to comment on. Also, the authors should apply correction procedures for multiple comparisons. This is the major part of my comments as it might impact on the conclusions.

6. PLOS authors have the option to publish the peer review history of their article (what does this mean?). If published, this will include your full peer review and any attached files.

Reviewer #1: Yes: Laurent Metzinger

Reviewer #2: No

---

## [Author Response · Author response to Decision Letter 0]

3 Apr 2020

Responses to the editor’s comments: 

Comment 1: Please ensure that your manuscript meets PLOS ONE’s style requirements, including those for file naming.

Reply 1: The manuscript and file naming had been adjusted according to PLOS ONE’s style requirements. 

Comment 2: In the ethics statement in the manuscript and in the online submission form, please provide additional information about the patient records/samples used in your retrospective study. Specifically, please ensure that you have discussed whether all data/samples were fully anonymized before you accessed them and/or whether the IRB or ethics committee waived the requirement for informed consent.

Reply 2: It had been corrected in the ethics statement on manuscript page 6, line 6 - 8. 

This retrospective population-based cohort study was approved by the Institutional Review Board (IRB) of Tri-Service General Hospital (IRB Registration Number: 2-105-05-082). All data are fully anonymized before we collected them and the IRB had waived the requirement for informed consent.

Comment 3: We noticed you have some minor occurrence(s) of overlapping text with the following previous publications(s), which needs to be addressed. In your revision ensure you cite all your sources (including your own works), and quote or rephrase any duplicated text outside the Methods section. Further consideration is dependent on these concerns being addressed. 

Reply 3: It had been rephrased and cited on page 6, line 11 - 17 and reference 6. 

Data from the NHIRD in Taiwan were used in this study [6]. The association between heat stroke and CKD events was investigated between 2000 to 2013 period. More than 99% of the population in Taiwan was covered by the National Health Insurance Program. The international Classification of Diseases, Ninth Revision (ICD-9) code was used for diagnosis [7]. The database in NHIRD also includes medications and patients’ demographics (such as socioeconomic status and residential area).

Comnet 4: Please include a separate caption for each figure in your manuscript.

Reply 4: Separate caption had been included in page 10, line 13; page 12, line 15 and 16. 

Response to the reviewers

Reviewer 1

Comment 1-1: Was CKD diagnosed according to eGFR? If yes, which equation was used to calculate it (CKD-EPI or other)?

Reply 1-1: It had been corrected on page 8, line 6 – 8.

Patients with CKD were divided into subgroups according to their eGFR, which was calculated according to MDRD (modification of diet in renal disease study) equation [8].

Comment 1-2: The paper would gain a lot of relevance if the associations were to be performed again according to the various CKD stages (1-5) or at least by subdividing patients in two groups, one for CKD stage 1 to 3a, the other from CKD stage 3b to 5.

Reply 1-2: It had been corrected on page 16, line 1 – 8 and page 16, table 5.

Furthermore, the effect of HI on the CKD subgroups were examined according to ICD 9 codes in Table 5. The effect of HI was statistically significant in those CKD stage 2-5 and CKD of unknown stage group. HI patients compared with control group showed increased CKD stage 2 risk (95% CI = 1.038-2.014, adjusted HR: 1.432, P = 0.029), CKD stage 3 risk (95% CI = 4.232-8.496, adjusted HR: 5.265, P < 0.001), CKD stage 4 risk (95% CI = 5.998-11.225, adjusted HR: 7.984, P < 0.001), and CKD stage 5 risk (95% CI = 9.137-20.986, adjusted HR: 11.106, P < 0.001) , respectively.

Comment 1-3: In figure 2, there seems to be an acceleration of CKD cases in the last 3 years in the HI group, can you please comment (heat wave in Taiwan or other explanation?)

Reply 1-3: It had been explained on page 12, line 10 – 13.

There seems to be an acceleration of CKD cases in the last 3 years in the HI group. It might be because of the annual average temperature was higher and there was more days with daily highest temperature > 35 oC in the last 3 years according to the database of Taiwan Central Weather Bureau.

Minor comment:

Abstract

Comment 1-4: Please correct “newly diagnosed with HI were followed”.

Reply 1-4: It had been corrected on page 3, line 10 and 11. 

We enrolled patients with HI who were followed in NHIRD system between 2000 and 2013.

Introduction

Comment 1-5: Heat is repeated three times in “Heat injury (HI) is the accumulation of heat resulting in the body’s inability to tolerate heat.”

Reply 1-5: It had been corrected on page 4, line 11.

 Heat injury (HI) is the accumulation of heat resulting in the body’s inability to tolerate it.

Rephrase

Comment 1-6: reductions in nephrons number, insufficient blood supply, cell cycle disruption, and disrupted repair mechanisms, which eventually CAUSE chronic kidney disease (CKD).

Reply 1-6: It had been corrected on page 5, line 9. 

However, sustained AKI can lead to renal interstitial fibrosis, reductions in nephron number, insufficient blood supply, cell cycle disruption, and disrupted repair mechanisms, which eventually cause chronic kidney disease (CKD).

Comment 1-7: Rephrase sentence “In addition to HI, other systemic co-mobidities were also examined in the multivariate analysis model to investigate whether HI is an independent risk factor for CKD, which develops in the majority of the population.”

Reply 1-7: It had been rephrased on page 5, line 18 and 19. 

In addition to HI, other systemic co-mobidities were also examined in the multivariate analysis model to investigate whether HI is an independent risk factor for CKD. , which develops in the majority of the population.

Comment 1-8: Separate “Materials and methodsData source”

Reply 1-8: It had been separated on page 6, line 3.

Results

Comment 1-9: separate “andcomorbidity”

Reply 1-9: It had been separated on page 10, line 16 and 17. 

and comorbidity

Comment 1-10: Several word corrections still appear in the text, for example “The risk of CKD is was” and “in the heat stroke and matched groups,” please correct.

Reply 1-10: It had been corrected on page 12, line 8 and page15, line 5.

The risk of CKD is was more pronounced and was higher after the fourth year of follow-up in the heat stroke group than in the control group.

We also investigated the risk of ESRD in the heat stroke and matched groups. ,

Comment 1-11: Please correct “Patients with heat stroke had higher CKD events”

Reply 1-11: It had been corrected on page 12, line 19. 

Patients with heat stroke had higher CKD events than the controls.

Comment 1-12: Discussion rephrase (Has and repetition if increase) “A recent study HAS shown that heat stress induced the increase in plasma lipopolysaccharide concentration and significant increases”

Reply 1-12: It had been rephrased on page 18, line 19 and 20. 

 A recent study have has shown that heat stress induced the increase in plasma lipopolysaccharide concentration, and significant increases in anti-inflammatory cytokines (IL-10 and IL-1ra ) and inflammatory responsive cytokines (IL-6)

Comment 1-13: “which are in concordance with the findings in previous studies”. Please cite them here

Reply 1-13: This sentence had been deleted on page 19, line 7 and 8. 

Our results demonstrate the higher risk of CKD after heat stroke, which are in concordance with the findings in previous studies.

Response to the reviewer 2:

Comment 2-1: Kindly put in the LINE numbers so it facilitates identification of the lines that the reviewer needs to comment on.

Reply 2-1: Line numbers had been put in the manuscript. 

Comment 2-2: Also, the authors should apply correction procedures for multiple comparisons. This is the major part of my comments as it might impact on the conclusions.

Reply 2-2: It had been corrected on page 9, line 18 and 19. 

The type I error due to multiple testing was corrected by the Bonferroni method. The P value < 0.001 was considered to be significant for multiple comparisons.

Comment 2-3: The P in P-value is not consistently expressed… Sometimes P, other times p or even p

Reply 2-3: It had been corrected in the manuscript page 9, line 15 and 17; page 15, line 9.

Comment 2-4: Materials and methodsData source 

 --space

Reply 2-4: It had been corrected on page 6, line3. 

Comment 2-5: The authors made no mention of correcting for multiple comparisons.

Reply 2-5: It had been corrected on page 9, line 18 and 19. 

The type I error due to multiple testing was corrected by the Bonferroni method. The P value < 0.001 was considered to be significant for multiple comparisons.

Comment 2-6: Indicate the limitations in Discussion with letters or numbers.

Reply 2-6: It had been corrected on page 20, line 11.

The strengths of our retrospective study included the national database derived from 1 one million sampled cases.

Materials and methods

Comment 2-7: The diagnostic codes were recorded according to the International Classification of Diseases, Ninth Revision (ICD-9), Reference

Reply 2-7: It had been corrected, on page 23, line 6 - 8.

Reference [7] International Classification of Diseases, Ninth Revision (ICD-9), Centers for Disease Control and Prevention, 1998, available from https://www.cdc.gov/nchs/icd/icd9.htm

Statistical analysis

Comment 2-8: The baseline demographic condition and comorbidities were compared between the study and control groups using the x2 test for categorical variables and the t-test for continuous variables [6]. 

The authors use descriptive statistics in the Results. I suggest they put in about descriptive statistics here. 

Reply 2-8: It had been corrected, on page 9, line 4. 

The baseline demographic condition and comorbidities were compared between the study and control groups using descriptive statistics. Chi-square test were used for categorical variables and t-test for continuous variables. 

Comment 2-9: In supplement Table 1, the median follow-up time of the heat stroke and control groups were 10.40 ± 13.70 and 10.97 ± 9.84 years, respectively

Median values look more like mean ± standard deviations. Median is usually followed range

Reply 2-9: It had been corrected, on page 16, line 15. 

In Table S1-1 and S1-2, the median mean follow-up time of the heat stroke and control groups were 10.40 ± 13.70 and 10.97 ± 9.84 years, respectively.

Comment 2-10: A recent study have shown that heat stress induced the increase in plasma…

 --- has

Reply 2-10: It had been rephrased on page 18, line 19. 

 A recent study have has shown that heat stress induced the increase in plasma lipopolysaccharide concentration,

Comment 2-11: Table 1. Characteristics of study in the baseline

 Specify the Location

Reply 2-11: It had been corrected on page 11, table 1. 

 Location in Taiwan 

Comment 2-12: Table 1-3. Use a corrective measure for the multiple P-values so that the authors minimize the risk of Type 1 error. 

Reply 2-12: It had been corrected on page 9, line 18 and 19.

The type I error due to multiple testing was corrected by the Bonferroni method. The P value < 0.001 was considered to be significant for multiple comparisons.

---

## [Decision Letter · Decision Letter 1]

30 Apr 2020

PONE-D-19-33292R1

Risk of chronic kidney disease in patients with heat injury: A nationwide longitudinal cohort study in Taiwan

PLOS ONE

Dear Dr CHU,

Thank you for submitting your manuscript to PLOS ONE. After careful consideration, we feel that it has merit but does not fully meet PLOS ONE’s publication criteria as it currently stands. Therefore, we invite you to submit a revised version of the manuscript that addresses the points raised during the review process.

The statistic issues are not resolved.

We would appreciate receiving your revised manuscript by 1st june. To enhance the reproducibility of your results, we recommend that if applicable you deposit your laboratory protocols in protocols.io, where a protocol can be assigned its own identifier (DOI) such that it can be cited independently in the future. For instructions see: http://journals.plos.org/plosone/s/submission-guidelines#loc-laboratory-protocols

We look forward to receiving your revised manuscript.

Kind regards,

Valérie Metzinger-Le Meuth, PhD

Academic Editor

PLOS ONE

Additional Editor Comments (if provided):

The authors should follow the recommendations of reviewers, especially on statistics issues.

Reviewers' comments:

Reviewer's Responses to Questions

**Comments to the Author**

1. If the authors have adequately addressed your comments raised in a previous round of review and you feel that this manuscript is now acceptable for publication, you may indicate that here to bypass the “Comments to the Author” section, enter your conflict of interest statement in the “Confidential to Editor” section, and submit your "Accept" recommendation.

Reviewer #1: All comments have been addressed

Reviewer #3: (No Response)

2. Is the manuscript technically sound, and do the data support the conclusions?

Reviewer #1: Yes

Reviewer #3: No

3. Has the statistical analysis been performed appropriately and rigorously? 

Reviewer #1: Yes

Reviewer #3: No

4. Have the authors made all data underlying the findings in their manuscript fully available?

Reviewer #1: Yes

Reviewer #3: Yes

5. Is the manuscript presented in an intelligible fashion and written in standard English?

Reviewer #1: Yes

Reviewer #3: No

6. Review Comments to the Author

Reviewer #1: changes are ok for me. blablablablablablablablablablablablablablablablablablablablablablablablablabla

Reviewer #3: The authors present results from a study of heat injury (HI) and development of chronic kidney disease (CKD), using data from the Taiwan National Health Insurance Research Database. They found a higher risk of CKD in those with heat injury compared to matched controls. Results were further explored by type of heat injury as well as by severity of CKD. The manuscript will be strengthened if the authors consider the following points.

1. Authors need to add some additional information about the Taiwan National Health Insurance Research Database. What years does it cover? What specifically does it capture? For example, it is not clear if data prior to 2000 are available for review, but authors selected 2000-2013 as the study period or if the database started in 2000. This is particularly important for understanding the control group who the authors describe as "never having HI"

2. Authors need to discuss the method used for matching (Figure 1 mentions propensity-score matched controls, but this approach is not mentioned anywhere in the methods). Related to the propensity score, authors are encouraged to include the model results used as the basis for propensity score matching as a supplemental table.

3. Authors mention 95% confidence intervals for incidence, but these are not presented anywhere (line 10, page 9)

4. Authors state in the methods that they used the Bonferroni method for handling multiple comparisons, but never mention it again. Authors should make note of findings that survive the correction (I'm guessing this might be the * in the tables, but that is not specified).

5. Table 1 has percentages for categories calculated out of the entire sample, even for the With and Without groups. Such a percentage makes sense for the Total column, but for the With and Without columns, such a percentage is difficult to interpret especially with the 1:4 matching. The percentages should be calculated out of the "n" for that column, so that when looking at the p-value, the reader can directly look at the column percentages to understand the differences (or lack thereof).

6. Table 1 - the p-values associated with the comorbidities are not correct based on the frequencies provided. When I run chi-square or Fisher's exact tests on the contingency tables, I get highly significant differences between the With and Without groups on the comorbidity variables.

7. Figure 2: authors report the numbers under the figure as No. at risk, but these correspond to the numbers with CKD. The number at risk is more informative. It isn't clear why there are counts of CKD given at 0 years, since all individuals are free of CKD at 0. Also, for the HI group, it is not clear why the numbers under the figure stay at 72 for years 11, 12, 13, while there are jumps in the figure, which should correspond to new cases.

8. Table 3: Is model 2 a single model with the various categories of HI used as predictors? The use of "stratified by variables" in the table title makes it unclear exactly what was done. Some of the categories have very low n (and a low number of events) which makes them uninformative. Also, 95% CIs should be included for the Adjusted HRs.

Minor points:

1. line 15 in Abstract: The authors use "13-year follow-up", but they do not have 13 years of follow-up for all individuals. Authors might consider using "13 year observation period" or something similar.

2. line 17 in Abstract: "higher CKD events" should be "an increased risk of CKD"

3. line 14 on page 4: "Heat stroke is type of" should be "Heat stroke is a type of"

4. line 15 on page 4: "accounts to 600" should be "accounts for 600"

5. lines 11-12 on page 7: authors mention that HI patients that could not be matched were excluded - these aren't reported in Figure 1 anywhere - how many were excluded due to lack of a match?

6. line 17 on page 7: though readers can likely figure it out, authors should define "index date"

7. line 18 on page 7: "All enrolled patients had been diagnosed with CKD" is confusing, since not everyone had CKD. I'm guessing the authors mean to say something similar to "Individuals were identified as having CKD if they had a diagnosis"

8. line 16 on page 8: "or after 31 December 2013" is confusing - do the authors just mean to say "31 December 2013" since that is the end of the observation period?

9. lines 5-6 on page 9: the sentence starting with "Multivariate Cox models" is confusing, since models were not generated simultaneously for the list of variables - those variables were included in all of the models.

10. lines 10-12 on page 9: the sentence starting with "The odds ratios of CKD" is awkwardly phrased and should be reworded.

11. Figure 1: "nuknown" is an incorrect spelling (box for Exclusion criteria)

12. line 12 on page 10: Authors mention no difference in insurance premiums, but this information is not presented in Table 1

13. Note under Table 1: "continue" should be "continuous"

14. lines 2-9 on page 12: the description of the curves may not be needed, as there are no statistical tests performed to support the various statements. The larger jumps in the later years could be due to smaller numbers of people still at risk.

15. Tables 4-5 do not need to include the results for the overall sample, since that is already provided in Table 2. They also have the exact same title, so authors should come up with more descriptive table titles.

16. Tables 4 and 5 also need some clarification - for example, is the row for "With HD" corresponding to a model where the outcome is CKD with HD vs no CKD? What does "Ratio" represent? Also, there are two columns labeled 95% CI - I'm assuming these are the lower limit and the upper limit for the CI, but that should be clarified. The sample sizes in the groups (for example "With HD" or "CKD Stage II") should be given in the tables.

17. Tables S-1 and S-2: "medium" should be "median". Also, how are the max follow-up times greater than 13 years if the study period is from Jan 1, 2000 to Dec 31, 2013?

18. line 11 on page 16: "duration of onset of HI and CKD" should be "average time between HI and onset of CKD" or something similar and "duration" on the next line should be changed similarly.

19. last sentence of the results: did the authors do a statistical test to support this statement?

20. line 3 on page 17: "was associated with" should be "had a"

21. line 8 on page 17: "temperatures" should be "temperature"

22. line 18 on page 17: is the New York study really based on 19.17 million patients?

23: line 14 on page 19: "strongly association" should be "strongly associated"

7. PLOS authors have the option to publish the peer review history of their article (what does this mean?). If published, this will include your full peer review and any attached files.

Reviewer #1: Yes: Laurent METZINGER

Reviewer #3: No

---

## [Author Response · Author response to Decision Letter 1]

31 May 2020

Response to the reviewer 3:

Comment 1. Authors need to add some additional information about the Taiwan National Health Insurance Research Database. What years does it cover? What specifically does it capture? For example, it is not clear if data prior to 2000 are available for review, but authors selected 2000-2013 as the study period or if the database started in 2000. This is particularly important for understanding the control group who the authors describe as "never having HI"

Response 1 : Thanks for the reviewer’s comment. Taiwan launched a single-payer National Health Insurance program on March 1, 1995. The database of this program contains registration files and original claim data for reimbursement. Large computerized databases derived from this system by the National Health Insurance Administration. We use data from 1995 to 1999 as wash-out period to make sure the cases enrolled from 2000 are new cases. 

Comment 2. Authors need to discuss the method used for matching (Figure 1 mentions propensity-score matched controls, but this approach is not mentioned anywhere in the methods). Related to the propensity score, authors are encouraged to include the model results used as the basis for propensity score matching as a supplemental table.

Response 2: Patients and controls were enrolled and propensity-score-matched (1:4) by age, sex, index date, comorbidities, and baseline medications.

Comment 3. Authors mention 95% confidence intervals for incidence, but these are not presented anywhere (line 10, page 9)

Response 3: This sentence had been deleted. 

Comment 4. Authors state in the methods that they used the Bonferroni method for handling multiple comparisons, but never mention it again. Authors should make note of findings that survive the correction (I'm guessing this might be the * in the tables, but that is not specified).

Response 4: It had been mentioned in the note of table 1. 

Comment 5. Table 1 has percentages for categories calculated out of the entire sample, even for the With and Without groups. Such a percentage makes sense for the Total column, but for the With and Without columns, such a percentage is difficult to interpret especially with the 1:4 matching. The percentages should be calculated out of the "n" for that column, so that when looking at the p-value, the reader can directly look at the column percentages to understand the differences (or lack thereof).

Response 5: The percentage of table 1 had been re-calculated according to the suggestion of reviewer. 

Comment 6. Table 1 - the p-values associated with the comorbidities are not correct based on the frequencies provided. When I run chi-square or Fisher's exact tests on the contingency tables, I get highly significant differences between the With and Without groups on the comorbidity variables.

Response 6: Some of the frequencies in table 1 was misplaced and miscalculated. It had been corrected, with red words and marked. 

Comment 7. Figure 2: authors report the numbers under the figure as No. at risk, but these correspond to the numbers with CKD. The number at risk is more informative. It isn't clear why there are counts of CKD given at 0 years, since all individuals are free of CKD at 0. Also, for the HI group, it is not clear why the numbers under the figure stay at 72 for years 11, 12, 13, while there are jumps in the figure, which should correspond to new cases.

Response 7: It is because the X-axis was not matched correctly with Y-axis. Figure 2 had been redrawn. 

Comment 8. Table 3: Is model 2 a single model with the various categories of HI used as predictors? The use of "stratified by variables" in the table title makes it unclear exactly what was done. Some of the categories have very low n (and a low number of events) which makes them uninformative. Also, 95% CIs should be included for the Adjusted HRs.

Response 8: Table 3 had been deleted from the manuscript. 

Minor points:

1. line 15 in Abstract: The authors use "13-year follow-up", but they do not have 13 years of follow-up for all individuals. Authors might consider using "13 year observation period" or something similar.

Response: It had been corrected on page 3, line 15.

2. line 17 in Abstract: "higher CKD events" should be "an increased risk of CKD"

Response: It had been corrected. 

3. line 14 on page 4: "Heat stroke is type of" should be "Heat stroke is a type of"

Response: It had been corrected. 

4. line 15 on page 4: "accounts to 600" should be "accounts for 600"

Response: It had been corrected.

5. lines 11-12 on page 7: authors mention that HI patients that could not be matched were excluded - these aren't reported in Figure 1 anywhere - how many were excluded due to lack of a match?

Response: This had been mentioned in new figure 1.

6. line 17 on page 7: though readers can likely figure it out, authors should define "index date"

Response: It had been defined. 

7. line 18 on page 7: "All enrolled patients had been diagnosed with CKD" is confusing, since not everyone had CKD. I'm guessing the authors mean to say something similar to "Individuals were identified as having CKD if they had a diagnosis"

Response: It had been rephrased. 

Individuals were identified as having CKD if they had a diagnosis should be confirmed…….

8. line 16 on page 8: "or after 31 December 2013" is confusing - do the authors just mean to say "31 December 2013" since that is the end of the observation period?

Response: It had been corrected.

or on 31 December 2013.

9. lines 5-6 on page 9: the sentence starting with "Multivariate Cox models" is confusing, since models were not generated simultaneously for the list of variables - those variables were included in all of the models.

Response: It had been corrected. 

10. lines 10-12 on page 9: the sentence starting with "The odds ratios of CKD" is awkwardly phrased and should be reworded.

Response: It had been rephrased. 

The odds ratios of CKD incidence rate after HI with and without adjustments for covariates with P < 0.05 in the univariate analysis was used for the conditional logistic regression analyses.

11. Figure 1: "nuknown" is an incorrect spelling (box for Exclusion criteria)

Response: It had been corrected. 

12. line 12 on page 10: Authors mention no difference in insurance premiums, but this information is not presented in Table 1

Response: It had been deleted. 

insurance premiums (in New Taiwan dollar $),

13. Note under Table 1: "continue" should be "continuous"

Response: It had been corrected.

14. lines 2-9 on page 12: the description of the curves may not be needed, as there are no statistical tests performed to support the various statements. The larger jumps in the later years could be due to smaller numbers of people still at risk.

Response: The description of the curves had been deleted. 

15. Tables 4-5 do not need to include the results for the overall sample, since that is already provided in Table 2. They also have the exact same title, so authors should come up with more descriptive table titles.

Response: The overall sample had been deleted. The table titles had been changed. 

16. Tables 4 and 5 also need some clarification - for example, is the row for "With HD" corresponding to a model where the outcome is CKD with HD vs no CKD? What does "Ratio" represent? Also, there are two columns labeled 95% CI - I'm assuming these are the lower limit and the upper limit for the CI, but that should be clarified. The sample sizes in the groups (for example "With HD" or "CKD Stage II") should be given in the tables.

Response: It had been corrected in table 4 and 5 (changed to table 3 and 4).

17. Tables S-1 and S-2: "medium" should be "median". Also, how are the max follow-up times greater than 13 years if the study period is from Jan 1, 2000 to Dec 31, 2013?

Response: It had been corrected in tables S-1 and S-2. The duration is 14 years from Jan 1, 2000 to Dec 31, 2013. 

(2000, 2001, 2002, 2003, 2004, 2005, 2006, 2007, 2008, 2009, 2010, 2011, 2012, 2013; totally 14 years.)

18. line 11 on page 16: "duration of onset of HI and CKD" should be "average time between HI and onset of CKD" or something similar and "duration" on the next line should be changed similarly.

Response: It had been corrected. 

19. last sentence of the results: did the authors do a statistical test to support this statement?

Response: The last sentence had been deleted. 

20. line 3 on page 17: "was associated with" should be "had a"

Response: It had been corrected. 

21. line 8 on page 17: "temperatures" should be "temperature"

Response: It had been corrected. 

22. line 18 on page 17: is the New York study really based on 19.17 million patients?

Response: The study population in that study included all residents of New York State. The population of New York State was 19.17 million. 

23: line 14 on page 19: "strongly association" should be "strongly associated"

Response: It had been corrected.

---

## [Decision Letter · Decision Letter 2]

19 Jun 2020

Risk of chronic kidney disease in patients with heat injury: A nationwide longitudinal cohort study in Taiwan

PONE-D-19-33292R2

Dear Dr. Chu,

We’re pleased to inform you that your manuscript has been judged scientifically suitable for publication and will be formally accepted for publication once it meets all outstanding technical requirements.

Kind regards,

Valérie Metzinger-Le Meuth, PhD

Academic Editor

PLOS ONE

Additional Editor Comments (optional):

The paper is accepted.

Reviewers' comments:

Reviewer's Responses to Questions

**Comments to the Author**

1. If the authors have adequately addressed your comments raised in a previous round of review and you feel that this manuscript is now acceptable for publication, you may indicate that here to bypass the “Comments to the Author” section, enter your conflict of interest statement in the “Confidential to Editor” section, and submit your "Accept" recommendation.

Reviewer #3: All comments have been addressed

2. Is the manuscript technically sound, and do the data support the conclusions?

Reviewer #3: (No Response)

3. Has the statistical analysis been performed appropriately and rigorously? 

Reviewer #3: (No Response)

4. Have the authors made all data underlying the findings in their manuscript fully available?

Reviewer #3: (No Response)

5. Is the manuscript presented in an intelligible fashion and written in standard English?

Reviewer #3: (No Response)

6. Review Comments to the Author

Reviewer #3: (No Response)

7. PLOS authors have the option to publish the peer review history of their article (what does this mean?). If published, this will include your full peer review and any attached files.

Reviewer #3: No

---

## [Editor Report · Acceptance letter]

24 Jun 2020

PONE-D-19-33292R2 

Risk of chronic kidney disease in patients with heat injury: A nationwide longitudinal cohort study in Taiwan 

Dear Dr. Chu:

I'm pleased to inform you that your manuscript has been deemed suitable for publication in PLOS ONE. Congratulations! Your manuscript is now with our production department. 

Kind regards, 

on behalf of

Dr. Valérie Metzinger-Le Meuth 

Academic Editor

PLOS ONE